# Progressive Platelet Rich Fibrin tissue regeneration matrix: Description of a novel, low cost and effective method for the treatment of chronic diabetic ulcers—Pilot study

Carlos José Saboia-Dantas[1], Paula Dechichi[1], Roberto Lichtsztejn Fech[2], Rafael Vilhena de Carvalho Furst[3], Rodrigo Daminello Raimundo[4]*, João Antonio Correa[3]

1 Laboratorio de Pesquisa em Reparo Tecidual, Universidade Federal de Uberlândia, Uberlândia, Minas Gerais, Brasil, 2 Instituto de Tecnologia e Ensino em Cirurgias, Uberlândia, Minas Gerais, Brazil, 3 Departamento de Cirurgia, Centro Universitário FMABC, Santo André, São Paulo, Brasil, 4 Laboratório de Delineamento de Pesquisas e Escrita Científica, Centro Universitário FMABC, Santo André, São Paulo, Brasil

* rodrigo.raimundo@fmabc.br

**Data Availability Statement:** All relevant data are within the paper and its Supporting information file.

## Abstract

### Introduction

Chronic lower limb ulcers (CLLU) are those injuries that persist for more than six weeks despite adequate care. They are relatively common; it is estimated that 10/1,000 people will develop CLLU in their lifetime. Diabetic ulcer, because of its unique pathophysiology (association between neuropathy, microangiopathy, and immune deficiency), is considered one of the most complex and difficult etiologies of CLLU for treatment. This treatment is complex, costly, and sometimes frustrating, as it is often ineffective, which worsens the quality of life of patients and makes its treatment a challenge.

### Objective

To describe a new method for treating diabetic CLLU and the initial results of using a new autologous tissue regeneration matrix.

### Method

This is a pilot, prospective, an interventional study that used a novel protocol of autologous tissue regeneration matrix for the treatment of diabetic CLLU.

### Results

Three male cases with a mean age of 54 years were included. A total of six Giant Pro PRF Membrane (GMPro) were used varying their application between one to three sessions during treatment. A total of 11 liquid phase infiltrations were performed varying their application

**Funding:** The authors received no specific funding for this work.

**Competing interests:** The authors have declared that no competing interests exist.

between three and four sessions. The patients were evaluated weekly and a reduction in the wound area and scar retraction was observed during the period studied.

## Conclusion

The new tissue regeneration matrix described is an effective and low-cost method for the treatment of chronic diabetic ulcers.

## 1 Introduction

Wound healing, in healthy individuals, is a highly precise and well-orchestrated process [1–3] whose capacity for tissue recovery, after damage, is given by the properties of the organisms that sustain the maintenance of the body's homeostasis.

The skin, being the largest organ by body surface area, is susceptible to injuries, either by mechanical damage, microbial infection, ultraviolet radiation, and extreme temperatures and its repair require the synchronization of several cell types and different molecules in sequential steps [4, 5], comprising four interconnected and overlapping phases: hemostasis, inflammation, proliferation, and remodeling phase [1]. The development of the entire process is essential to re-establish the skin's protective barrier. Interruptions or defects in these delicate phases can lead to a chronic wound state [6]. Failures in this process affect some 40 million patients with chronic wounds worldwide, with an impact on healthcare systems [1, 3, 4].

Diabetes, vascular disease, and aging, as well as genetic variations involved in the tissue repair process, are the main factors contributing to chronic wounds, causing diabetic foot ulcers, vascular ulcers (venous and arterial), pressure ulcers, or complex wounds, leading to long-term sequelae [7, 8]. After an initial unwanted process in that process, leading to wound chronification. In fact, after an initial insult, wounds that do not undergo a healing process within a period of 06 weeks are considered chronic wounds and have as characteristic, a prolonged inflammatory phase, which prevents the dermal and epidermal cells to respond to chemical agents.

Numerous therapies, both systemic and local, have been proposed for the treatment of chronic wounds. Systemic administration of antibiotics, even in high doses, is inferior to the antisepsis method, because antibiotics do not adequately penetrate the wound biofilms [9, 10]. The use of antibodies, peptides, hormones, amino acids, or derivatives, has a positive impact on chronic wounds, but despite these results, general systemic administration is limited by difficulties in selectively targeting the tissue and the emergence of possible adverse effects [1, 11, 12, 14]. Local therapies, such as physical and pharmacological treatment, are the most common choices [1]. Other options, such as skin grafts and dressing therapies using non-cellular films and bases (decellularized matrices, porous biopolymer bases, nanofiber meshes, and hydrogels), are available. Although there is a wide range of possible interventions, many of these treatments are only moderately effective, expensive, or even experimental, and there is a need for new and more effective therapies [4].

The use of hemoderivatives has been used for the past 50 years to treat skin wounds [8, 13, 14] and is one of the oldest approaches to regenerative medicine.

Currently, many Platelets Rich Plasma (PRP) methods are still commercialized and find applications in the treatment of skin wounds [15]. PRP represents the first generation of blood concentrates, but the complexity of obtaining it [15, 16] and the cost of application per treatment session make this technology inaccessible to most patients, however, PRP offers benefits

for the treatment of ulcers refractory wounds, as well as extensive and complex wounds [15], is a safe and effective modality to improve wound healing and can improve the quality of life of these patients [17]. A concept of broad versatility has been attributed to platelets, in addition to hemostatic activities, as indispensable structures for numerous physiological responses, including angiogenesis, inflammation, innate immunity and wound healing [9]. In tissue repair, platelets initiate the process of blood clot formation by associating with leukocytes and surrounding cells to form a temporary tissue, which will be replaced by the native tissues of the wound bed, or scar tissue. In addition to the cells, a three-dimensional and functional extracellular matrix must be present, functioning as a strong structure. With PRP, such conditions do not exist, limiting its use in the treatment of extensive and complex wounds when used alone [16, 18], however, this can be mitigated with the use in combination with a fibrin or another type of matrix [17].

In the early 2000s, Choukroun et al proposed changes in the methodology of obtaining blood concentrates [16, 19]. Thus, the second generation, represented by Leukocyte and Platelet Rich Fibrin (L-PRF), appeared, simplifying the obtaining of blood concentrates [19, 20], besides improving the mechanical and biological characteristics for clinical use [18–21]. With the advent of the PRF concept, it has become possible to treat refractory ulcers using membranes created by compressing PRF clots obtained in glass tubes. This method offers a very interesting, easy-to-use, low-cost, and extremely effective therapeutic option for millions of individuals, allowing wound closure without adverse events if standard therapy fails [15, 22]. Currently, second-generation blood concentrates have several protocols based on variations in centrifugation time [23, 24], aiming to improve cellularity [25], the conductive potential of concentrates, favoring histogenesis and angiogenesis [26], in addition to their application in injectable form l for tissue biostimulation [27–29], modulation of the inflammatory response [30] and biofunctionalization of biomaterials [31–33]. In 2019, a modification of the PRF protocol was proposed and verified by Saboia-Dantas and Dechichi [10], in the Tissue Regeneration Research Laboratory (LAPERT) at the Federal University of Uberlândia (FUU), called Progressive Platelet Rich Fibrin (PRO-PRF). The objective was to improve membrane strength, and cell distribution and facilitate the application of this blood concentrate in wound treatment and tissue reconstruction. After being well characterized by histological (optical, transmission, and scanning electron microscopy) and mechanical analyses (Fig 1a–1d),

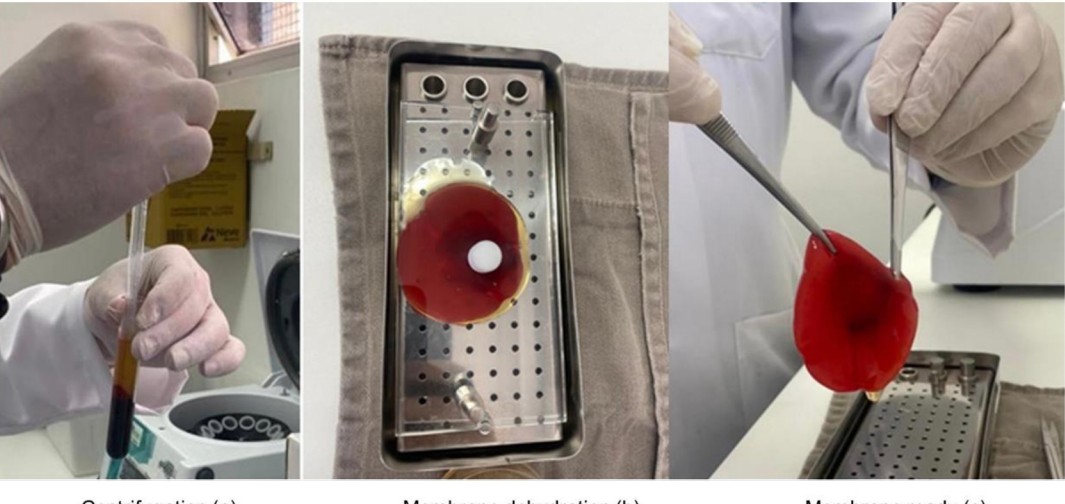

Centrifugation (a)    Membrane dehydration (b)    Membrane ready (c)

**Fig 1. Membrane preparation—(a) Centrifugation; (b) Membrane dehydration and (c) Membrane ready.**

PRO-PRF has been satisfactorily used in the treatment of intraoral and cutaneous wounds [34]. In this paper, the aim was to describe an innovative, feasible, and cost-effective approach for the treatment of chronic wounds of diabetic etiology and the initial results of using this autologous tissue regeneration matrix.

## 2 Method

The study was carried out at the vascular surgery outpatient clinic of *Centro Hospitalar do Município de Santo André* (CHMSA). This is a pilot, prospective, interventional study, approved by the Research Ethics Committee of the *Centro Universitário FMABC* (#5.346.616). All patients participating in the research agreed (written and witnessed) and signed the informed consent form (ICF) and authorization term for image use (TAUI). Patients over the age of 18 years, with chronic lower limb ulcer (CLLU) of diabetic etiology, without macroangiopathic ischemic disease, without active infection after debridement, without limitations in regards to area, depth or type of tissue exposed, and with sociocognitive ability to follow the treatment and care protocol were included in this study. Patients with ulcers of other etiologies were excluded.

### 2.1 Wound management protocol

**2.1.1 Wound preparation.** Cleaning and antisepsis of the chronic wounds were performed with an aqueous chlorhexidine solution before the application of the blood concentrates.

**2.1.2 Preparation of blood concentrate (PRO-PRF).** Blood samples were obtained by venipuncture, using Collection needles (scalp) and plastic tubes (PET), without blood collection additives (VACUETTE®, Grainer, Americana, São Paulo, Brazil). Ten to 12 tubes were collected, depending on the extent of the wound, and immediately transferred to a fixed-angle rotor centrifuge (DT 4000 Daiki Lab Bran Digital Centrifuge), with a total capacity of 12 tubes, and centrifuged to produce the PRO-PRF blood concentrate.

We used a new protocol developed by Saboia-Dantas and Dechichi [10], in the Tissue Regeneration Research Laboratory (LAPERT) at the Federal University of Uberlândia (FUU).

The blood samples were centrifuged for a total time of 15 minutes, with a progressive increase in speed (RPM—revolutions per minute) in intervals of 05 min (700 RPM / 1300 RPM / 2400 RPM) (Fig 1).

After centrifugation, the blood concentration obtained in each tube was aspirated with a 10 ml hypodermic syringe and a 16G needle. The volume of PRO-PRF obtained was deposited in a receptacle for the modeling of a PRO-PRF clot, and then pressed in a stainless-steel box with a glass plate (PRF Box) for serum drainage and formation of a Giant PRO PRF membrane (GMPro) (Fig 1). In addition, part of this volume was aspirated and inoculated into the wound bed.

**2.1.3 PRO-PRF application and wound protection.** The PRO-PRF, still in the gel phase, was immediately inoculated at the edges of the CLLU and deposited on the wound surface. In the subsequent step, the PRO-PRF membrane was implanted into the wound bed and fixed and stabilized using cyanoacrylate adhesive (Fig 2).

Finally, the wound was covered and wrapped with multi-pored polyvinyl chloride (PVC) film for transudate drainage and secondary dressing with gauze and crepe bandage to absorb fluids (Fig 2).

The PVC cover remained on for 7 days and the secondary dressing was changed daily. The patients were followed up in the outpatient clinic weekly. At the beginning of the treatment, as well as in all subsequent patient visits, the lesions were photographed, the wound dimensions

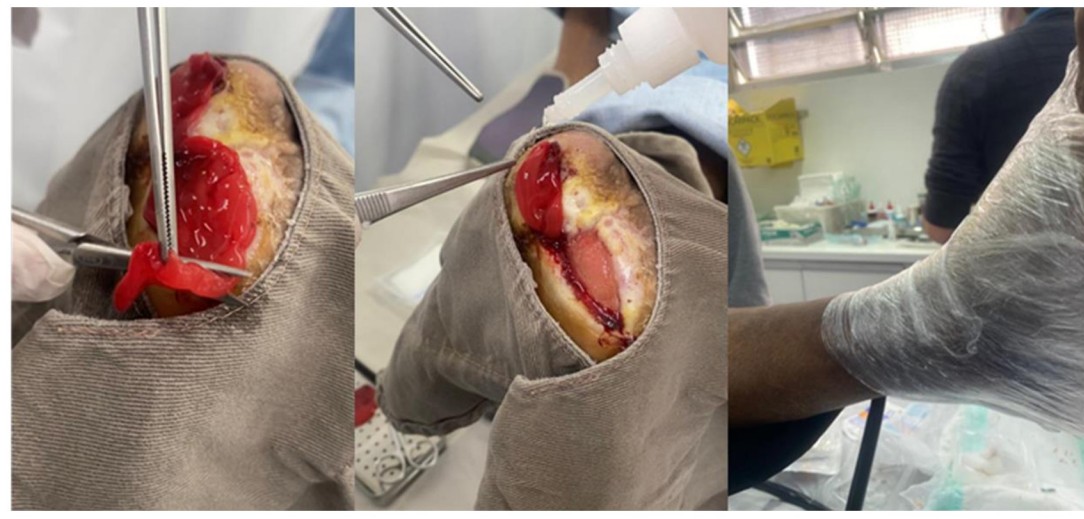

Customization (a)   Cyanoacrylate fixation (b)   Bandage with PVC (c)

**Fig 2. Membrane implant—Customization (a); Cyanoacrylate fixation (b) and Bandage with PVC (c).**

were measured, local signs of infection and, depending on the evolution of the wound, a new application of PRO-PRF or conventional dressing with caprylic/capric triglyceride (typically sourced from coconut oil or sunflower) oil was chosen.

The protocol described in this peer-reviewed article is published on protocols.io, https://dx.doi.org/10.17504/protocols.io.n2bvj8wqpgk5/v1.

## 3 Results

Three cases, all male, were included in this pilot study. Their ages ranged from 51 to 57 years, with a mean of 54 years. They were treated in a period from May 2022 to August 2022. They had type II diabetes mellitus with diagnoses ranging from five to 16 years. All patients in the sample had chronic diabetic ulcers that were difficult to heal for a period ranging from 12 to 32 weeks. In this sample, all patients had a single wound, with a total of three ulcers. The ulcers were without infection and were previously treated with local care, systemic antibiotic therapy, and photodynamic therapy ranging from 2 to 4 sessions. The topography and dimensions of the initial lesions were right fourth polydactyl and right forefoot with an area of 37 cm$^2$, left hallux amputation with an area of 15 cm$^2$ and left metatarsal dorsal aspect of the foot with an area of 24 cm$^2$. The treatment was performed over a period ranging from two to four weeks with an average treatment time of three weeks (Table 1). None of these three cases presented were offloading wound, however, this is not a contraindication of the technique.

All wounds were treated with an autologous tissue regeneration matrix combining giant membranes (GMPro) with its injectable liquid form, both obtained by progressive centrifugation and following the described protocol.

Blood was collected without interruption and multiple venipuncture attempts. A blood volume ranging from 32 milliliters to 96 milliliters was collected with an average of 64 milliliters.

The membranes (GMPro) formed in a time that ranged from 10 minutes to 20 minutes with an average formation time of 15 minutes.

A total of 7 membranes (GMPro) were used varying their application between 2 and 4 sessions with an average of 2 applications during the treatment. A total of 11 liquid phase

**Table 1. Description of the sample cases.**

| | |
|---|---|
| Case 1 | A 51-year-old male patient with type II diabetes for 16 years was admitted with an infected diabetic foot. He was submitted to surgical treatment with debridement of devitalized tissues and fasciotomy in the lateral and plantar surfaces of the distal third of the left foot, he was also treated with systemic antibiotic therapy and four sessions of photodynamic therapy (PDT), after the infection was eliminated, he started the proposed treatment with a 24 cm$^2$ ulcer, he was submitted to the implantation of three membranes (GMPro) and four ProPrf infiltrations in the liquid phase, with a 12-week follow-up. |
| Case 2 | A 55-year-old male patient with type II diabetes for 12 years was admitted with an infected diabetic foot. He was submitted to surgical treatment with amputation of the left hallux, debridement of devitalized tissues, and ample fasciotomy in the plantar surface, he was also treated with systemic antibiotic therapy and two sessions of photodynamic therapy (PDT), after the infection was eliminated, he started the proposed treatment with a 15 cm$^2$ ulcer, he was submitted to the implantation of two membranes (GMPro) and three ProPrf infiltrations in the liquid phase, with a 12-week follow-up. |
| Case 3 | A 57-year-old male patient with type II diabetes for five years was admitted with an infected diabetic foot. He was submitted to surgical treatment with amputation of the fourth polydactyl of the left foot, debridement of devitalized tissues, and ample fasciotomy on the dorsal and plantar surfaces of the foot; he was also treated with systemic antibiotic therapy and three sessions of photodynamic therapy (PDT); after the infection was eliminated, he started the proposed treatment with a 37 cm$^2$ ulcer; he was submitted to the implantation of two membranes (GMPro) and four PRO-PRF infiltrations in the liquid phase, with a 12-week follow-up. |

infiltrations were performed varying their application between three and four sessions with an average of 3.66 applications during the treatment.

The patients were followed up weekly and the dimensions (area and depth) of the wounds and local aspects related to infection or other complications were observed.

After the first session the patients were reevaluated weekly and a reduction in the area of the wounds was observed. This reduction ranged from 20.4% to 54% with an average reduction of 36.6% of their area. After the second week, the reduction ranged from 17.6% to 25.6% with an average reduction of 22%. After the third week, this reduction ranged from 32% to 46% with an average reduction of 39%.

All patients in this pilot study showed complete healing of the lesion. This healing occurred between a period ranging from seven to 12 weeks with an average of nine weeks for healing.

During this period, there was no case of local infection or systemic complication. In the three cases studied, there was a reduction in the perimeter of the lesion and significant scar retraction with the implant and application of PRO-PRF, and there was no need for hospitalization or amputation of the limb during the study period. The cases reported are shown in Figs 3–5.

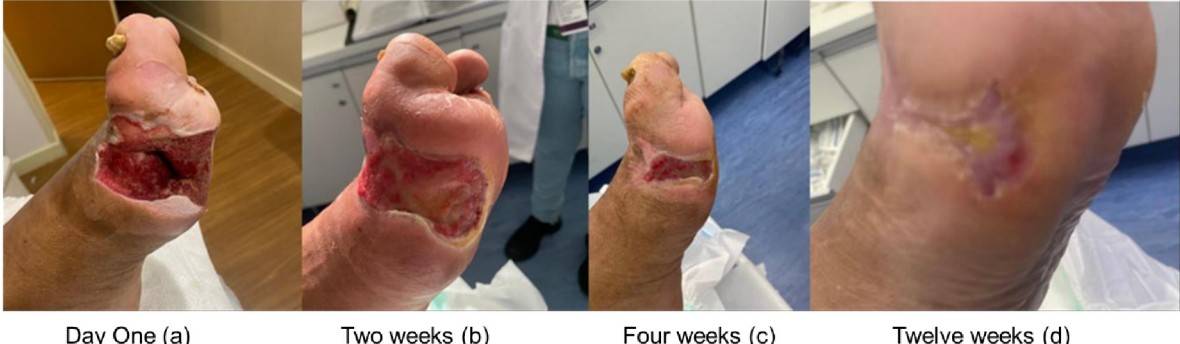

Day One (a)          Two weeks (b)          Four weeks (c)          Twelve weeks (d)

**Fig 3. Follow-up of wound evolution (Case 1).** Evolution of the case 1: Day One (a); Two weeks (b); Four weeks (c) and Twelve weeks (d).

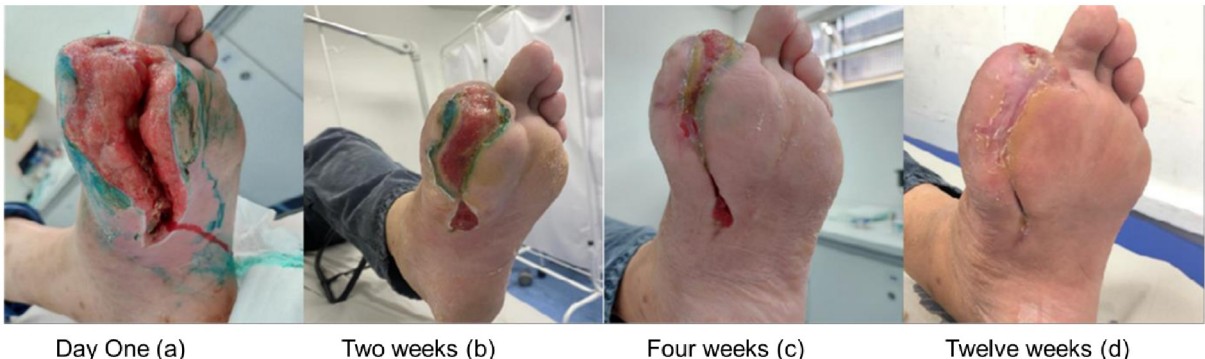

Day One (a)     Two weeks (b)     Four weeks (c)     Twelve weeks (d)

**Fig 4. Follow-up of wound evolution (Case 2).** Evolution of the case 2: Day One (a); Two weeks (b); Four weeks (c) and Twelve weeks (d).

## 4 Discussion

Only in the 19th century did the antiseptic technique emerge as a breakthrough for acute and chronic wound healing. As early as the 20th century, modern options for wound healing management emerged, and among them were approaches focused on tissue engineering. Currently, advances in the field of biomaterials and a deeper understanding of the wound-healing process have given rise to a large number of new therapies and strategies [1]. Unfortunately, most of them are costly to be widely used in all patients and are not always guaranteed to be effective.

Skin wounds, especially if they are recalcitrant despite proper care, have a dramatic impact on the quality of life, productivity, and life expectancy of patients. Non-healing status is associated with high treatment costs and is a major cause of morbidity leading to loss of function [15] and mortality for patients after 5 years of amputation is approximately 50% [8]. Thus, it is not surprising that in recent decades, appropriate and well-planned management of chronic wounds has become important [3].

Among the various therapies proposed for chronic wound healing, based on the principles of tissue engineering, we can mention the local application of blood concentrates. The work of Pinto et al [8] demonstrated a significant potential of L-PRF for wound healing without adverse events, signifying a low-cost and extremely efficient strategy in the management of chronic wounds. Furthermore, when L-PRF was employed, recurrence was not observed

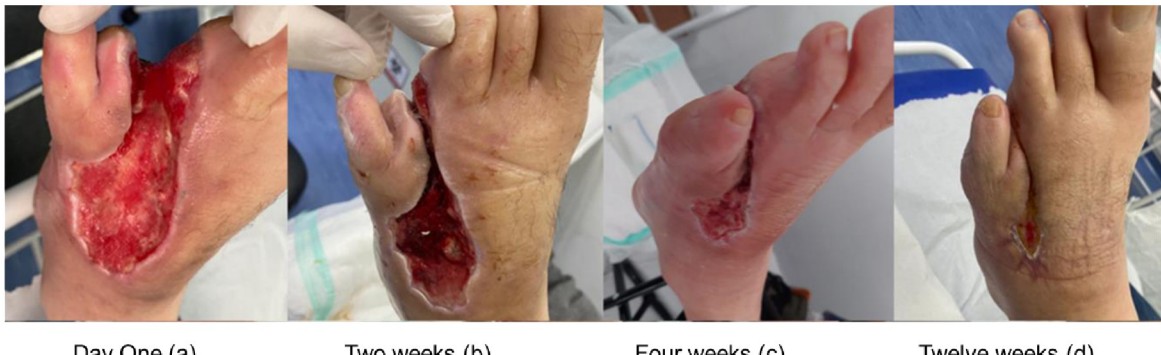

Day One (a)     Two weeks (b)     Four weeks (c)     Twelve weeks (d)

**Fig 5. Follow-up of wound evolution of case 3.** Evolution of the case 3: Day One (a); Two weeks (b); Four weeks (c) and Twelve weeks (d).

in the first year of follow-up, indicating that it is not only a good promoter of wound closure, but helps to achieve better quality regenerated tissue [15]. The extracellular matrix provides a bioactive environment that controls the behavior of cells using chemical and mechanical signals. Components of the extracellular matrix are actively involved in wound healing because of their ability to influence cell behavior (proliferation, adhesion, and migration), differentiation, and cell death through integrins, cytokines, and growth factors. It actively participates in the regulation of growth factors, receptors, hydration level, and pH of the local tissue environment [6]. Similarly, in the case of the establishment of injuries, the blood clot constitutes a temporary immune defense and repair tissue. Its cellular and molecular elements assume, until the proliferative phase, the role of the cells and the extracellular matrix of the injured tissue.

Considering their structure and biological properties [16, 20–23], blood concentrates are optimized clots [15], accelerating the repair process and improving the microenvironment conditions for tissue restructuring.

Since 2013, our group has used second-generation membranes, mainly L-PRF and A-PRF, as temporary fillers and natural autologous dressing [10]. In 2019, we started using third-generation giant membranes (GMPro) and injectable PRO-PRF with excellent clinical results in intraoral wound regeneration, and recently in chronic skin wound healing, with similar success as reported by Pinto et al [8]. However, this was the first time that GMPro, and the injectable form were used in combination in the treatment of chronic wounds on diabetic feet. The application of GMPro to the wound, initially stabilized using a cyanoacrylate adhesive and held in tensile equilibrium, favors the closure of extensive and complex wounds. The positive effect seems to be due to the immediate restoration of the cellular matrix, constituting a temporary and replaceable base with proven inductive and conductive action [21–25]. The injectable form and the membranes in association modulate the inflammatory response and confer immunological protection to the wound [23], with studies demonstrating the antimicrobial potential of PRF [27–29].

Snyder et al (2020) conducted a study to analyze the cost-effectiveness of five advanced skin substitutes in the treatment of foot ulcers in patients with diabetes. The number of applications was also 12 weeks of treatment and the total cost ranged from $1245 to $7647. Our pilot study had a total cost (complete treatment) between $200 and $800 (depending on the number of sessions), which represents a low cost by Brazilian and international standards, even more so when compared to other regeneration matrices and industrialized dressings [33].

In our clinical experience, the use of PRO-PRF in the treatment of wounds on the oral mucosa and skin, both in the form of GMPro and injectable, has demonstrated the acceleration of the repair process, favoring angiogenesis, and in general, histogenesis, correcting deficiencies commonly presented by chronic wounds at the cellular and molecular level. This new approach is a simple, safe, and low-cost option using an autologous tissue regeneration matrix easily obtained from blood and should be considered a relevant therapeutic option for the treatment of chronic diabetic ulcers. Further studies to elucidate the local phenomena involved in the repair process of these wounds should be performed, to better understand the interaction of third-generation blood concentrates with the tissues.

## 5 Conclusion

In our preliminary study, the use of PRO-PRF in the form of GMPro and injections of its liquid phase proved to be an easy-to-perform, low-cost, and effective therapeutic option in the treatment of chronic cutaneous wounds in diabetic feet.

## Supporting information

**S1 File.**
(PDF)

## Author Contributions

**Conceptualization:** Carlos José Saboia-Dantas, Paula Dechichi, Roberto Lichtsztejn Fech, Rafael Vilhena de Carvalho Furst, João Antonio Correa.

**Data curation:** Carlos José Saboia-Dantas, Paula Dechichi, Roberto Lichtsztejn Fech, Rafael Vilhena de Carvalho Furst, João Antonio Correa.

**Formal analysis:** Carlos José Saboia-Dantas, Paula Dechichi, Roberto Lichtsztejn Fech, Rodrigo Daminello Raimundo, João Antonio Correa.

**Investigation:** Carlos José Saboia-Dantas, Roberto Lichtsztejn Fech, Rafael Vilhena de Carvalho Furst, João Antonio Correa.

**Methodology:** Carlos José Saboia-Dantas, Paula Dechichi, Roberto Lichtsztejn Fech, Rafael Vilhena de Carvalho Furst, Rodrigo Daminello Raimundo, João Antonio Correa.

**Project administration:** Carlos José Saboia-Dantas, Roberto Lichtsztejn Fech, Rafael Vilhena de Carvalho Furst, João Antonio Correa.

**Supervision:** Carlos José Saboia-Dantas, Paula Dechichi, Roberto Lichtsztejn Fech, Rafael Vilhena de Carvalho Furst, João Antonio Correa.

**Validation:** Carlos José Saboia-Dantas, Paula Dechichi, Roberto Lichtsztejn Fech, Rafael Vilhena de Carvalho Furst, Rodrigo Daminello Raimundo, João Antonio Correa.

**Visualization:** Rafael Vilhena de Carvalho Furst, Rodrigo Daminello Raimundo, João Antonio Correa.

**Writing – original draft:** Carlos José Saboia-Dantas, Paula Dechichi, Roberto Lichtsztejn Fech, Rafael Vilhena de Carvalho Furst, Rodrigo Daminello Raimundo, João Antonio Correa.

**Writing – review & editing:** Carlos José Saboia-Dantas, Paula Dechichi, Roberto Lichtsztejn Fech, Rafael Vilhena de Carvalho Furst, Rodrigo Daminello Raimundo, João Antonio Correa.

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
