## [Decision Letter · Decision Letter 0]

6 Feb 2023

PONE-D-22-32724Progressive Platelet Rich Fibrin tissue regeneration matrix: Description of a novel, low cost and effective method for the treatment of chronic diabetic ulcers - Pilot studyPLOS ONE

Dear Dr. Raimundo,

Thank you for submitting your manuscript to PLOS ONE. After careful consideration, we feel that it has merit but does not fully meet PLOS ONE’s publication criteria as it currently stands. Therefore, we invite you to submit a revised version of the manuscript that addresses the points raised during the review process.

We look forward to receiving your revised manuscript.

Kind regards,

Prabhakar Orsu, PhD

Academic Editor

PLOS ONE

Journal Requirements:

Reviewers' comments:

Reviewer's Responses to Questions

**Comments to the Author**

1. Does the manuscript report a protocol which is of utility to the research community and adds value to the published literature?

Reviewer #1: Yes

2. Has the protocol been described in sufficient detail?

To answer this question, please click the link to protocols.io in the Materials and Methods section of the manuscript (if a link has been provided) or consult the step-by-step protocol in the Supporting Information files.

The step-by-step protocol should contain sufficient detail for another researcher to be able to reproduce all experiments and analyses.

Reviewer #1: Yes

3. Does the protocol describe a validated method?

Reviewer #1: Yes

4. If the manuscript contains new data, have the authors made this data fully available?

Reviewer #1: Yes

**5. Is the article presented in an intelligible fashion and written in standard English?**

Reviewer #1: Yes

6. Review Comments to the Author

Reviewer #1: The authors present a small case series in which patients with diabetic complex foot wounds were healed using a new type of second-generation PRF. I have just some minor comments to add to this well-written study.

1. Introduction: “…but the complexity of obtaining them (15,15) a…” Was there another reference to be inserted?

2. Introduction: “In addition, PRP offers limited benefits for the treatment of refractory ulcers as well as extensive and complex wounds (15).” I would say that is a little misleading. See for example: Meznerics FA, Fehérvári P, Dembrovszky F, Kovács KD, Kemény LV, Csupor D, Hegyi P, Bánvölgyi A. Platelet-Rich Plasma in Chronic Wound Management: A Systematic Review and Meta-Analysis of Randomized Clinical Trials. J Clin Med. 2022 Dec 19;11(24):7532.

3. Introduction: “In addition to the cells, a three-dimensional, functional extracellular matrix must be present, functioning as a strong framework. With PRP, such conditions do not exist, limiting its use in the treatment of extensive and complex wounds (16,17).” It depends on whether it’s used in combination with a fibrin or other type of matrix. That should be mentioned.

4. Methods: UCMI. Please define this acronym. Please also check that all acronyms are spelled out at first usage.

5. Inclusion criteria: Were there any limitations in regard to area, depth, or type of tissue exposed?

6. Methods: Did the wounds need debridement after the initial treatment?

7. Methods: “The PRO-PRF, still in the fluid phase…” I might suggest gel phase because if the term fluid is used it suggests a liquid to most clinicians.

8. Methods: “At the beginning of the treatment, as well as in all the returns, the lesions were photographed, the wound dimensions were measured, local signs of infection and, depending on the evolution of the wound, a new application of PRO-PRF or conventional dressing with sunflower oil was chosen.” I might suggest: “At the beginning of the treatment, as well as in all subsequent patient visits…” Is the sunflower oil dressing a commercial product (if so, please state). If not, please describe how it was made.

9. Did any of the wounds need offloading (i.e., they were anatomically plantar wounds?

10. Results: I would suggest to use only one decimal place.

11. Discussion: “Our pilot study had a cost between $200 and $800.” Do these values represent the lowest and highest costs? What exchange rate (US dollars vs Brazilian Reals) was used?

7. PLOS authors have the option to publish the peer review history of their article (what does this mean?). If published, this will include your full peer review and any attached files.

Reviewer #1: **Yes: **Marissa Janine Carter

---

## [Author Response · Author response to Decision Letter 0]

20 Mar 2023

Dear Reviewer

We very much appreciate you for your highly constructive reviews to our submission. We have revised the material to eliminate the issues raised. The added or modified words, phrases, and sentences are in red. We hope the present version can be accepted.

Reviewers' comments:

Reviewer's Responses to Questions

Comments to the Author

1. Does the manuscript report a protocol which is of utility to the research community and adds value to the published literature?

Reviewer #1: Yes

Answer: Thank you 

2. Has the protocol been described in sufficient detail?

To answer this question, please click the link to protocols.io in the Materials and Methods section of the manuscript (if a link has been provided) or consult the step-by-step protocol in the Supporting Information files.

The step-by-step protocol should contain sufficient detail for another researcher to be able to reproduce all experiments and analyses.

Reviewer #1: Yes

Answer: Thank you 

3. Does the protocol describe a validated method?

Reviewer #1: Yes

Answer: Thank you 

4. If the manuscript contains new data, have the authors made this data fully available?

Reviewer #1: Yes

Answer: Thank you 

5. Is the article presented in an intelligible fashion and written in standard English?

Reviewer #1: Yes

Answer: Thank you 

6. Review Comments to the Author

Reviewer #1: The authors present a small case series in which patients with diabetic complex foot wounds were healed using a new type of second-generation PRF. I have just some minor comments to add to this well-written study.

1. Introduction: “…but the complexity of obtaining them (15,15) a…” Was there another reference to be inserted?

Answer: The reference has been corrected.

“…but the complexity of obtaining them (15,16) a…”

2. Introduction: “In addition, PRP offers limited benefits for the treatment of refractory ulcers as well as extensive and complex wounds (15).” I would say that is a little misleading. See for example: Meznerics FA, Fehérvári P, Dembrovszky F, Kovács KD, Kemény LV, Csupor D, Hegyi P, Bánvölgyi A. Platelet-Rich Plasma in Chronic Wound Management: A Systematic Review and Meta-Analysis of Randomized Clinical Trials. J Clin Med. 2022 Dec 19;11(24):7532.

Answer: Thanks for the comment. The sentence has been rewritten and the reference has been included.

“Currently, many Platelets Rich Plasma (PRP) methods are still commercialized and find applications in the treatment of skin wounds (15). PRP represents the first generation of blood concentrates, but the complexity of obtaining it (15,16) and the cost of application per treatment session make this technology inaccessible to most patients, however, PRP offers benefits for the treatment of ulcers refractory wounds, as well as extensive and complex wounds (15), is a safe and effective modality to improve wound healing and can improve the quality of life of these patients (17). A concept of broad versatility has been attributed to platelets, in addition to hemostatic activities, as indispensable structures for numerous physiological responses, including angiogenesis, inflammation, innate immunity and wound healing (9).”

3. Introduction: “In addition to the cells, a three-dimensional, functional extracellular matrix must be present, functioning as a strong framework. With PRP, such conditions do not exist, limiting its use in the treatment of extensive and complex wounds (16,17).” It depends on whether it’s used in combination with a fibrin or other type of matrix. That should be mentioned.

Answer: Thanks for the comment. The sentence has been rewritten.

In addition to the cells, a three-dimensional and functional extracellular matrix must be present, functioning as a strong structure. With PRP, such conditions do not exist, limiting its use in the treatment of extensive and complex wounds when used alone (16,18), however, this can be mitigated with the use in combination with a fibrin or another type of matrix (17).

4. Methods: UCMI. Please define this acronym. Please also check that all acronyms are spelled out at first usage.

Answer: It was a typing error, thanks.

“Patients over the age of 18 years, with chronic lower limb ulcer (CLLU) of diabetic etiology…”

5. Inclusion criteria: Were there any limitations in regard to area, depth, or type of tissue exposed? 6. Methods: Did the wounds need debridement after the initial treatment?

Answer: Thanks for the comment. The sentence has been rewritten.

“Patients over the age of 18 years, with chronic lower limb ulcer (CLLU) of diabetic etiology, without macroangiopathic ischemic disease, without active infection after debridement, without limitations in regard to area, depth or type of tissue exposed, and with sociocognitive ability to follow the treatment and care protocol were included in this study. Patients with ulcers of other etiologies were excluded.”

7. Methods: “The PRO-PRF, still in the fluid phase…” I might suggest gel phase because if the term fluid is used it suggests a liquid to most clinicians.

Answer: Thanks for the comment. The term has been changed

“The PRO-PRF, still in the gel phase, was immediately inoculated at the edges of the CLLU and deposited on the wound surface. In the subsequent step, the PRO-PRF membrane was implanted into the wound bed and fixed and stabilized using cyanoacrylate adhesive (Figure 2).”

8. Methods: “At the beginning of the treatment, as well as in all the returns, the lesions were photographed, the wound dimensions were measured, local signs of infection and, depending on the evolution of the wound, a new application of PRO-PRF or conventional dressing with sunflower oil was chosen.” I might suggest: “At the beginning of the treatment, as well as in all subsequent patient visits…” 

Answer: Thanks for the comment. The term has been changed.

“The PVC cover remained on for 7 days and the secondary dressing was changed daily. The patients were followed up in the outpatient clinic weekly. At the beginning of the treatment, as well as in all subsequent patient visits,…”

Is the sunflower oil dressing a commercial product (if so, please state). If not, please describe how it was made.

Answer: Thanks for the comment. The description was made.

“The PVC cover remained on for 7 days and the secondary dressing was changed daily. The patients were followed up in the outpatient clinic weekly. At the beginning of the treatment, as well as in all subsequent patient visits, the lesions were photographed, the wound dimensions were measured, local signs of infection and, depending on the evolution of the wound, a new application of PRO-PRF or conventional dressing with caprylic/capric triglyceride (typically sourced from coconut oil or sunflower) oil was chosen.”

9. Did any of the wounds need offloading (i.e., they were anatomically plantar wounds?

Answer: Thanks for the comment. The sentence has been rewritten.

Three cases, all male, were included in this pilot study. Their ages ranged from 51 to 57 years, with a mean of 54 years. They were treated in a period from May 2022 to August 2022. They had type II diabetes mellitus with diagnoses ranging from five to 16 years. All patients in the sample had chronic diabetic ulcers that were difficult to heal for a period ranging from 12 to 32 weeks. In this sample, all patients had a single wound, with a total of three ulcers. The ulcers were without infection and were previously treated with local care, systemic antibiotic therapy, and photodynamic therapy ranging from 2 to 4 sessions. The topography and dimensions of the initial lesions were right fourth polydactyl and right forefoot with an area of 37 cm2, left hallux amputation with an area of 15 cm2 and left metatarsal dorsal aspect of the foot with an area of 24 cm2. The treatment was performed over a period ranging from two to four weeks with an average treatment time of three weeks (Table 1). None of these three cases presented were offloading wound, however, this is not a contraindication of the technique.

10. Results: I would suggest to use only one decimal place.

Answer: The manuscript has been revised and the decimal places have been changed

“…15 cm2 and left metatarsal dorsal aspect of the foot with an area of 24 cm2”

“… after the infection was eliminated, he started the proposed treatment with a 24 cm2 ulcer,..”

“…After the first session the patients were reevaluated weekly and a reduction in the area of the wounds was observed. This reduction ranged from 20.4% to 54% with an average reduction of 36.6% of their area. After the second week, the reduction ranged from 17.6% to 25.6% with an average reduction of 22%. After the third week, this reduction ranged from 32% to 46% with an average reduction of 39%.”

11. Discussion: “Our pilot study had a cost between $200 and $800.” Do these values represent the lowest and highest costs? What exchange rate (US dollars vs Brazilian Reals) was used?

Answer: We appreciate the comments. The discussion was reviewed by the authors

Snyder and Ead (2020) conducted a study to analyze the cost-effectiveness of five advanced skin substitutes in the treatment of foot ulcers in patients with diabetes. The number of applications was also 12 weeks of treatment and the total cost ranged from $1245 to $7647. Our pilot study had a total cost (complete treatment) between $200 and $800 (depending on the number of sessions), which represents a low cost by Brazilian and international standards, even more so when compared to other regeneration matrices and industrialized dressings (33).

Do you want your identity to be public for this peer review? For information about this choice, including consent withdrawal, please see our Privacy Policy.

Reviewer #1: Yes: Marissa Janine Carter

Answer: Dear Marissa Janine Carter. We very much appreciate you for your highly constructive reviews to our submission. We have revised the material to eliminate the issues raised. We hope the present version can be accepted.

---

## [Editor Report · Decision Letter 1]

6 Apr 2023

Progressive Platelet Rich Fibrin tissue regeneration matrix: Description of a novel, low cost and effective method for the treatment of chronic diabetic ulcers - Pilot study

PONE-D-22-32724R1

Dear Dr. Raimundo,

We’re pleased to inform you that your manuscript has been judged scientifically suitable for publication and will be formally accepted for publication once it meets all outstanding technical requirements.

Kind regards,

Prabhakar Orsu, PhD

Academic Editor

PLOS ONE
---

## [Editor Report · Acceptance letter]

25 Apr 2023

PONE-D-22-32724R1 

Progressive Platelet Rich Fibrin tissue regeneration matrix: Description of a novel, low cost and effective method for the treatment of chronic diabetic ulcers - Pilot study 

Dear Dr. Raimundo:

I'm pleased to inform you that your manuscript has been deemed suitable for publication in PLOS ONE. Congratulations! Your manuscript is now with our production department. 

Kind regards, 

on behalf of

Dr. Prabhakar Orsu 

Academic Editor

PLOS ONE